# Subnanometer-resolution structure determination in situ by hybrid subtomogram averaging - single particle cryo-EM

Ricardo M. Sanchez[1,2,4], Yingyi Zhang [1,2,4], Wenbo Chen[1,2], Lea Dietrich[3] & Mikhail Kudryashev [1,2]✉

Cryo-electron tomography combined with subtomogram averaging (StA) has yielded high-resolution structures of macromolecules in their native context. However, high-resolution StA is not commonplace due to beam-induced sample drift, images with poor signal-to-noise ratios (SNR), challenges in CTF correction, and limited particle number. Here we address these issues by collecting tilt series with a higher electron dose at the zero-degree tilt. Particles of interest are then located within reconstructed tomograms, processed by conventional StA, and then re-extracted from the high-dose images in 2D. Single particle analysis tools are then applied to refine the 2D particle alignment and generate a reconstruction. Use of our hybrid StA (hStA) workflow improved the resolution for tobacco mosaic virus from 7.2 to 4.4 Å and for the ion channel RyR1 in crowded native membranes from 12.9 to 9.1 Å. These resolution gains make hStA a promising approach for other StA projects aimed at achieving subnanometer resolution.

[1] Max Planck Institute for Biophysics, Max-von-Laue Strasse, 3, 60348 Frankfurt am Main, Germany. [2] Buchmann Institute for Molecular Life Sciences, Goethe University of Frankfurt am Main, Max-von-Laue Strasse, 15, 60348 Frankfurt am Main, Germany. [3] Department of Structural Biology, Max Planck Institute for Biophysics, Max-von-Laue Strasse, 3, 60348 Frankfurt am Main, Germany. [4] These authors contributed equally: Ricardo M. Sanchez, Yingyi Zhang. ✉email: misha.kudryashev@biophys.mpg.de

Cryo-electron microscopy (cryo-EM) is a versatile technique for achieving precise structural knowledge of biological processes over a range of scales. Single-particle cryo-EM (SPA) is capable of yielding high-resolution structures of purified proteins of 1.62 Å[1] and higher, with the average resolution of the maps, deposited in the Electron Microscopy Data Bank (EMDB) in 2019, hitting 5.6 Å. In contrast, the average resolution of the StA structures deposited in 2019 is ~28 Å, although a few structures have been reported in the range of 3−4 Å resolution[2–5]. StA structures at subnanometer resolution are of great interest as they allow resolving secondary structural elements and provide unique insights into the function of proteins in their native context[6–9].

The conventional StA workflow[10] starts by (1) recording dose-fractionated movies at each of the projection angles in a tilt series, (2) aligning each of the motion-corrected micrographs, or tilt images, in the series, typically using gold fiducials, (3) estimating the defocus and performing contrast transfer function (CTF) correction on the aligned stacks by phase flipping, and (4) generating tomographic reconstructions using the CTF-corrected aligned stacks. Next, the reconstructed tomograms are used for (5) particle picking and extraction followed by (6) alignment of the particles to a common average with optional classification. Additional steps may include 3D CTF correction[4,11,12], correction for local sample deformation based on gold fiducial alignment[13], and/or constrained refinement of tomographic geometry based on the positions of the particles[5,14]. Conventional StA workflow faces several bottlenecks, which make achieving high resolution challenging.

First, beam-induced specimen movement is not uniform[15], and may surpass 10 Å[16,17]. As a result, tomograms are not recorded from rigid objects, but rather from ones that are continuously changing as the electron dose accumulates. Second, due to the distribution of the typical total electron dose of 60−150 e−/Å² over 40−60 projections, the precision of defocus determination is lower than in single-particle cryo-EM. Third, the effective thickness of the sample increases during tilting and introduces an additional defocus gradient in the direction of the electron beam. This leads to further difficulties in defocus determination and CTF correction. CTF correction error of 250 nm limits the maximum achievable resolution to ~10 Å, on a 300-kV microscope[18]. Finally, a large number of particles is required to be collected and processed, in order to achieve high resolution. Recording tomograms at lower magnification provides higher throughput; however, this comes at a price of data quality[10] and increased anisotropic magnification distortions[19]. Unless magnification anisotropy is precisely quantified and corrected for the used magnification, a distortion of up to 2.7% would correspond to up to tens of pixels in a 4k micrograph in a direction arbitrarily oriented to the tilting axis. Tomographic reconstruction and subsequent subtomogram averaging from such data will suffer from non-precise registration reducing the attainable resolution.

Here we present a workflow that combines the advantages of single-particle cryo-EM and StA with a focus on achieving subnanometer resolution. Our workflow consists of recording a higher dose micrograph at zero-degree tilt followed by recording the remainder of the tilt series in the standard way, incorporating the "high-dose" image into the tilt series and performing StA using the described above workflow. At the last step the particles extracted in 2D from the "high-dose" images are locally refined against their average. We refer to the tomograms recorded using this method as hybrid tomograms and the entire workflow as hybrid subtomogram averaging (hStA). We evaluate the hStA workflow on tomograms of tobacco mosaic virus (TMV) recorded at a relatively low magnification of an electron microscope corresponding to 2.2 Å/pixel in counted mode, with an aim to collect a larger particle number. We further compare the performance of conventional StA processing of tomograms recorded with the standard dose-symmetric tilt scheme[20] against conventional StA of hybrid tomograms. Finally, we evaluate the performance of hStA on an ion channel RyR1 preserved in native sarcoplasmic reticulum (SR) membranes isolated from rabbit skeletal muscle. Our results show significant improvement in resolution by using hStA compared to the conventional StA.

## Results

**Hybrid StA-SPA workflow**. We extended the dose-symmetric SerialEM script[20] to record a "high-dose" movie as the first exposure of the tilt series (scripts are available in Supplementary Notes 1 and 2). The electron dose for this "high-dose" movie is 15 e−/Å², with the total dose for the entire tilt series equal to approximately 95 e−/Å² (Fig. 1a). Anisotropic motion correction is performed on the "high-dose" micrograph using MotionCor2[21] with the last frame used as the reference. The movies for the remaining tilts are aligned globally. The micrographs are normalized to have the same mean values; their standard deviations are adjusted according to the electron dose applied for each micrograph with the "high-dose" image resulting in lower standard deviation. We assumed the following empirical relationship: $(\sigma_{HD})^2/(\sigma_{LD})^2 = e^-_{LD}/e^-_{HD}$. For our data collection scheme, with $e^-_{HD} = 15\,e^-/Å^2$ and $e^-_{LD} = 2\,e^-/Å^2$, the ratio of standard deviations is $\sigma_{HD} = 0.37 \times \sigma_{LD}$. The tilt series are then aligned using gold beads and tomograms are reconstructed conventionally using Imod[22], the particles are identified in the tomograms and StA is performed using Dynamo[23] (Fig. 1b). Based on the final StA alignment, the particles are located in the original "high-dose" image, and extracted in 2D. The rotations needed to align these 2D particles to the average are calculated as a combination of the geometrical transforms from the StA workflow (Fig. 1b). In our implementation, these rotations are stored in a widely used STAR-formatted file in the ZYZ Euler angles representation. The defocus difference relative to the centre of the tomogram is known for each particle, based on the Z-heights of the particle in the tomogram. The updated defocus values are also stored in the STAR file. We refer to the reconstruction resulting from the "high-dose" micrographs as a hybrid map. The positions and rotations of the particles are further refined using this hybrid map as a reference, with the resulting reconstruction referred to as the refined map. During this refinement step, using Relion 3.0[24], only small local searches are performed to the translations and rotations of each particle. The processing script and the usage instructions can be found in Supplementary Note 3.

**Resolution gain with hStA**. In order to assess the benefits of our approach experimentally, we first used the highly stable and helically symmetric TMV as a test sample. We acquired tomograms on a Titan Krios (Thermo Fisher Scientific) equipped with a K2 camera (Gatan) and energy filter at an intermediate nominal magnification of 64,000× (1.1 Å/pixel in superresolution mode) with the aim of imaging larger fields of view and therefore more particles. We recorded small datasets of conventional dose-symmetric tomograms (4570 asymmetric units) and "hybrid tomograms" (4190 asymmetric units), with an equivalent total electron dose of approximately 95 e−/Å². As expected, the "high-dose" micrographs resulted in more visible and better defined Thon rings (Fig. 2a, b), as well as lower *B*-factors. Our calculations showed that when high frequencies are present in micrographs and are used for fitting of CTF, defocus can be detected with higher precision (Fig. 2c, see "Methods").

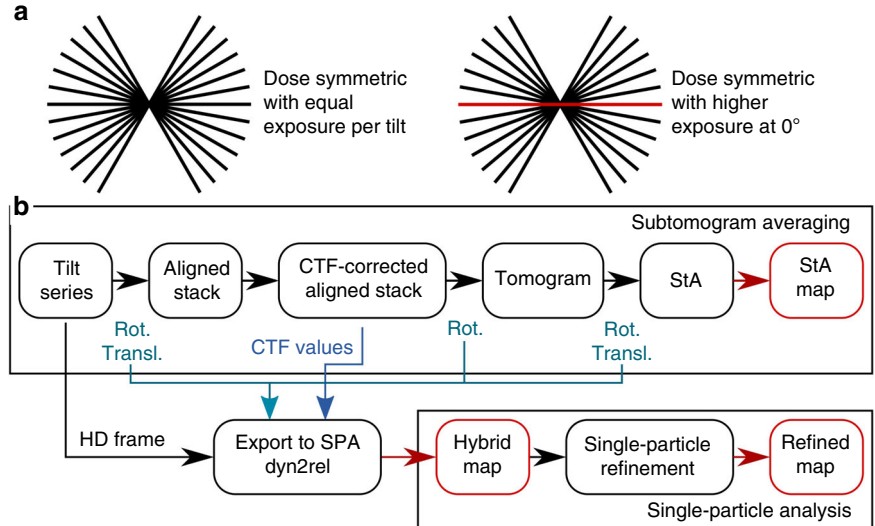

**Fig. 1 Scheme of the Hybrid StA-SPA workflow. a** A schematic depicting the distribution of dose in the dose-symmetric scheme[20] with uniform dose distribution (left) and with an increased exposure for the untilted image shown in red (right panel). **b** Data flow diagram for conventional and hybrid StA, black boxes correspond to the intermediate steps, red boxes indicate the output maps, blue arrows indicate geometric transforms performed between the processing steps and the CTF information which are exported for hybrid StA processing.

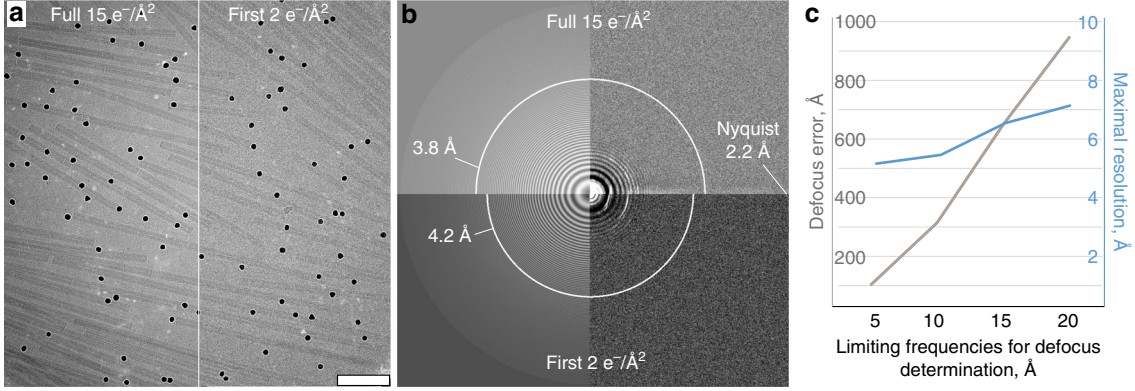

**Fig. 2 Improved CTF determination for the Hybrid StA. a** A representative micrograph of a 15 e$^-$/Å$^2$, 0° tilt image (left) shown together with the first 2 e$^-$/Å$^2$ of the same image (right). Scale bar: 100 nm. **b** The detected Thon rings for the two images in (**a**). The white arcs represent the maximum resolution reported by Gctf[38]. **c** Relationship between the maximum frequencies used for defocus determination (X-axis) and the precision of defocus estimation (Y-axis). Grey line: error in defocus determination with limited frequencies compared to the defocus determination with the full range of frequencies; blue line: maximal resolution up to which the estimated CTF fits with the data, as reported by Gctf[38].

We next compared the performance of the conventional StA data processing on hybrid and conventional tomograms of the described TMV datasets. The processing was done using the conventional StA workflow (in Dynamo, see "Methods") with the identical parameters and helical symmetry applied. The resulting reconstructions had similar resolutions of 9.8 and 10.0 Å respectively (Fig. 3a–c). The similarity in resolution for the subtomogram averages resulting from the conventional and hybrid tomograms suggests that collecting data using our hybrid method does not detriment data processing performed by the conventional StA workflow.

Next, using the same settings, we recorded a larger dataset of hybrid tomograms of TMV, containing 20,214 asymmetric units. Conventional StA processing with helical symmetry resulted in a 7.2 Å structure (Fig. 3d). The Hybrid map, obtained using data only from the "high-dose" image, resulted in a 7.5 Å structure (Fig. 3e). Interestingly, conventional subtomogram averages had "halos" adjacent to observed protein density (marked with red

asterisk in Fig. 3a, b, d). To the knowledge of the authors, these "halos" are present to a certain extent in all structures produced by subtomogram averaging. Similar effect has been reported when CTF correction on 2D images has been performed by phase flipping, while the application of an adjusted Wiener filter recovered the original images more precisely[25]. Conventional StA processing bases CTF correction on phase flipping[26], while in hStA Wiener filtration (implemented in Relion) is employed; as a result, while the FSC curves for the conventional StA map and the hybrid map look similar, the hybrid map has a better appearance (Fig. 3e). Finally, we used Relion to refine the hybrid map, which improved the resolution to 5.2 Å, clearly showing alpha-helical secondary structure. Subsequent per-particle refinement of the defocus values resulted in a reconstruction at 4.4 Å (Fig. 3f, g), reaching the counted sampling limit.

We next probed if deteriorating the quality of tomographic data resulting in misalignments and reduced resolution could be recovered by hybrid refinement. For this we perturbed the

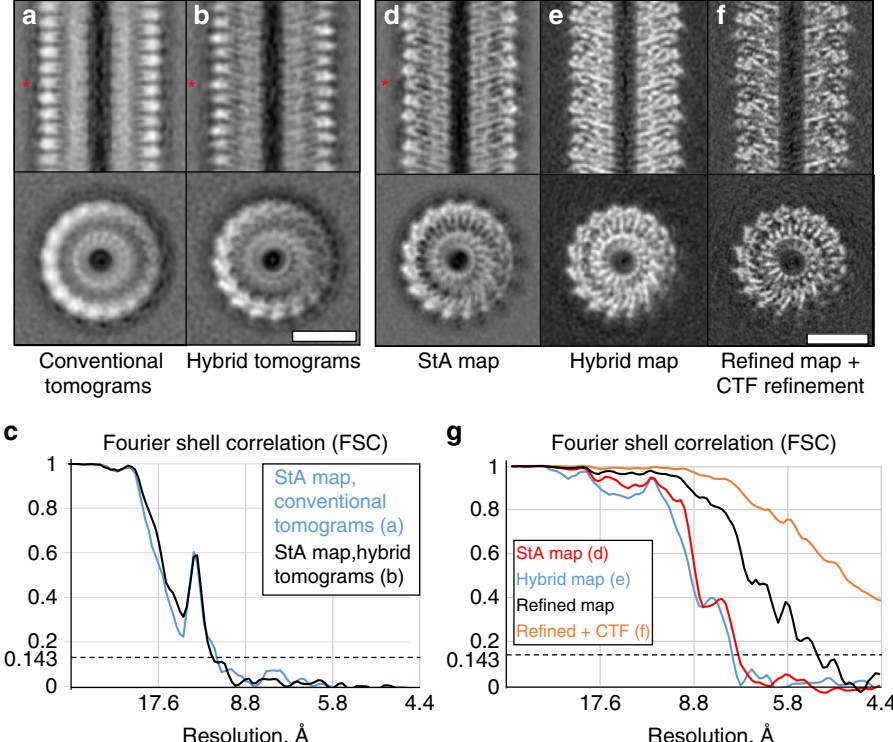

**Fig. 3 Resolution improvement by the application of *hStA* to TMV. a–c** Structures of TMV (**a**, **b**) and the corresponding FSC curves (**c**) as processed by the conventional subtomogram averaging workflow. The dataset in **a** was recorded using a conventional dose-symmetric scheme and in **b** using the proposed hybrid scheme. Note the "halos" observed next to the density in **e**, **f** and marked with the asterisks. **d–g** Processing of a larger dataset of hybrid tomograms of TMV: **d** conventional StA map; **e** Hybrid map, reconstruction only from the high-dose image; **f** and the refined map with the refined CTF; **g** the corresponding FSC curves. Scale bars: 10 nm.

translations and rotations of subtomogram alignment for the TMV dataset by a small, moderate and significant amounts (see "Methods"). The resolution of the maps generated with misalignment reduced from 7.2 Å to 8.8, 11, and 14 Å respectively. Application of hybrid refinement enabled us to recover similar high resolution for all the cases (Supplementary Fig. 1).

**Structure of an ion channel RyR1 in native membranes.** We next applied hStA to the structural analysis of ryanodine receptor RyR1 in native SR vesicles purified from fresh rabbit skeletal muscle. We previously reported a structure of RyR1 that reached 12.6 Å from 2547 particles with C4 symmetry applied[27] by conventional StA. Here, we recorded a set of hybrid tomograms of the same samples at a higher nominal magnification of 81,000×, corresponding to a pixel size of 1.7 Å (Fig. 4a, Supplementary Movie 1). We picked 2715 particles and used Dynamo to obtain an StA map of 12.9 Å resolution (Fig. 4b), using C4 symmetry. We then exported the particles and performed refinement in Relion applying C4 symmetry. Presence of side-, top- and intermediate views of the protein (Supplementary Fig. 2) together with C4 symmetry allowed complete sampling of the orientations. The resulting refined map from 2563 particles had a resolution of 9.1 Å (Fig. 4c, d, Supplementary Table 1), and allowed direct observation of secondary structure in the transmembrane and cytoplasmic domains (Fig. 4e). The reconstruction converged to a higher-resolution structure only if local searches were allowed; discarding the angular information obtained from StA and attempting to determine the angles de novo did not lead to high-resolution reconstructions.

## Discussion

Here we report a hybrid workflow for subtomogram averaging that redistributes the electron dose across a tilt series, allowing to leverage the image processing tools developed for SPA and StA to achieve significantly higher resolution. We demonstrate that, by using the hStA approach, subnanometer-resolution maps can be obtained with data acquired at intermediate magnifications of electron microscopes. We attribute the observed gain in resolution to four main factors: first, by redistributing more of the available electron dose to the "high-dose" image, the defocus estimation, and therefore the CTF correction can be performed more precisely. Importantly, better CTF correction is performed on the data with higher SNR which is used for the final reconstruction. Second, knowing the height of each particle within its tomogram allows to account for the defocus gradient within the sample during CTF correction. Third, the ability to compensate for imperfections in tomographic geometry and for beam-induced sample movement in the final refinement step. Finally, the use of single-particle-analysis software for reconstruction enables application of Wiener filtering to improve the density quality. In some cases, such as out TMV dataset, further refinements like per-particle defocus refinement could lead to further resolution improvements.

The distribution of the total electron dose over the tilt angles is driven by the total dose tolerable by the sample and the amount of signal needed for alignment of subtomograms. In our test datasets we limited the total applied electron dose aiming to preserve the well-defined ~11 Å signal of TMV, which is significantly reduced after the exposure of ~100 e−/Å[28]. We allocated 2 e−/Å² per each of the 40 projections and used the remaining ~15% of the total electron dose for the "high-dose"

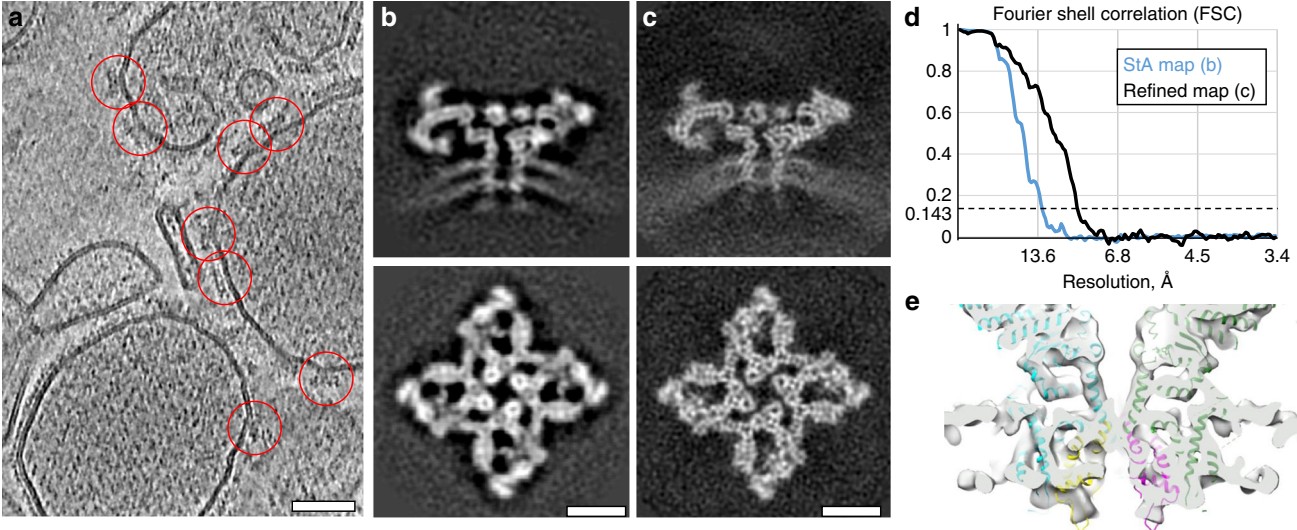

**Fig. 4 Structure of RyR1 in native membranes at subnanometer resolution. a** A slice through a tomogram of native SR vesicles extracted from rabbit skeletal muscle. Red circles highlight RyR1 particles on the membrane of SR vesicles. Scale bar: 50 nm. **b** Slices through a subtomogram average generated from 2715 particles at a resolution of 12.9 Å. **c** Slices though a refined map at 9.1 Å resolution generated from 2563 particles. Scale bars in **b**, **c**: 10 nm. **d** Corresponding FSC curves between the respective independently processed half-sets. **e** A volume rendered visualization of the transmembrane domain of the refined map of RyR1 with a rigidly fit atomic model (PDB ID: 5TB2 from reference[43]).

micrograph. However, for the samples tolerating a higher total dose, or requiring less projections for alignment, the exposure for the high-dose image could be further increased as it is used for the final alignment of particles and the 3D reconstruction. For the RyR1 dataset we showed that the hybrid refinement could be performed on micrographs of 16 e⁻/Å², which increased the resolution from 12.9 Å to subnanometer 9.1 Å using 2563 RyR1 particles and the application of C4 symmetry. The relatively small number of particles was probably the limiting factor for reaching higher resolution. The increase of the particle number and of the electron dose for the untilted images provides a potential for further resolution improvement. In both the examined cases of TMV and RyR1, we could achieve fully sampled orientations. Cases of strong preferred orientation and lack of symmetry will result in anisotropy of resolution of the final maps. In such cases hybrid tomography could be performed with high-dose images acquired at a tilted stage[29] and the data processing could be done by hStA as described (Supplementary Notes 2 and 3).

Three-dimensional reconstructions from high-dose 2D projections directed by StA have been pioneered by Bharat et al.[30]. There, helical parameters of individual tubes, made of the retroviral protein Gag, were determined by StA and this information was used for helical reconstruction from photographic film. Recently, Song et al.[31] presented Tygress, a general implementation of 3D reconstruction from high-dose images guided by tomography. While the ideas of Tygress and hStA are similar, differences exist. First, Tygress does not use the untilted high-dose projection for tomogram generation or subtomogram averaging. In our implementation we introduce a normalization procedure that enables all the collected data to be used for subtomogram alignment. Secondly, the implementation details vary: both methods are implemented in MATLAB, in Tygress, a Peet[32] project is refined with Frealign[33] inside Matlab, using a set of configuration files and a GUI to execute the final refinement step. Our approach uses a minimal set of Matlab functions and classes to export a Dynamo project into a STAR file for refinement in Relion[24] or other packages. Both implementations provide a significant improvement in resolution: the use of Tygress allowed obtaining a reconstruction of the outer doublet microtubules from an intact *Tetrahymena thermophila* ciliary axoneme at

10.6 Å from 112,386 particles. Using hStA we reached 9.1 Å using a much lower number of 10,252 asymmetric units; although the differences in the numbers of necessary particles could be partially attributed to the other factors such as ice thickness, differences in hardware.

The optimal scheme for tomographic data collection is still under debate—while the dose-symmetric tilt scheme[20] provides better data, it comes at the price of lower throughput[3]. In our implementation of hStA only the first "high-dose" image is used for the final reconstruction, which relaxes the quality requirements for the remaining projections in the tilt series. Furthermore, hybrid tomographic acquisition and processing may be naturally combined with collecting fast tilt series[34,35] to increase throughput. If combined with conservatively increasing the angular step and/or reducing the angular coverage of the tilt series, throughput could further increase to one tomogram every few minutes, improving the current throughput by tenfold. Furthermore, we envisage that hStA could be beneficial for fiducial-less samples, such as FIB-milled lamella[36], as the lack of fiducials for tilt series alignment reduces the quality of the resulting tomograms, affecting the quality of the resulting subtomogram averages. Based on our results, we believe that use of hybrid tomograms and application of our hStA workflow would be beneficial for a large number of projects aimed at obtaining structures of molecular complexes with subnanometer resolution or higher.

## Methods

**Sample preparation and collection of cryo-electron tomograms**. The TMV sample, 33 mg/mL was gently mixed with 10 nm protein-A gold nanoparticles (Cell Microscopy Core, Netherlands, UMC Utrecht), with a volume ratio of 1−0.8, immediately before plunge-freezing. Three microlitres of sample mixed with gold fiducials was then applied on a twice glow discharged (PELCO easiGlow™ Glow Discharge Cleaning System) Quantifoil Au 1.2/1.3 300-mesh grid, blotted using a Mark IV Vitrobot, and plunge-frozen in liquid ethane. Grids were stored in liquid nitrogen until data collection. Tomograms were recorded using two transmission electron microscopes, a first-generation FEI Titan Krios and an FEI Titan Krios G2 (Thermo Fisher Scientific) operated at 300 kV, with the specimen maintained at liquid nitrogen temperatures. Tomograms were acquired at a nominal magnification of 64,000× (2.2 Å per pixel in counting mode) using SerialEM with the dose-symmetric script[20] and the hybrid variant (Supplementary Note 1). Movie stacks with 0.3 e⁻/Å² per frame (1.5−1.8 e⁻/Å² per projection) were recorded on a

post-GIF K2 Summit direct electron detector (Gatan, Inc.) in superresolution mode. The total electron dose for both the conventional dose-symmetric and hybrid tomograms was kept the same, ~95 e$^-$/Å$^2$. The conventional tomograms had the total dose equally distributed over 41 tilt images, from −60 to 60° with a tilting step of 3°. The hybrid tomograms had a zero-tilt image with a total dose of 15−20 e$^-$/Å$^2$, with the remaining dose equally distributed over the remaining 40 tilt images over the same angular range.

For RyR1 data collection, the same preparations as in ref. [27] were used. For sample preparation, 60 g of fresh rabbit skeletal muscle tissue from the hind leg and back was ground using a meat grinder, and homogenized using a blender with 300 mL of homogenization buffer (0.5 mM ethylenediaminetetraacetic acid (EDTA), 10% sucrose, 20 mM $Na_4O_7P_2$, 20 mM $NaH_2PO_4$, and 1 mM $MgCl_2$, pH 7.1) with addition of the following protease inhibitors: 2.6 µg/mL aprotinin, 1.4 µg/mL pepstatin, and 10 µg/mL leupeptin. Homogenates derived from a total of 180 g of muscle were centrifuged in a Beckman Coulter rotor JLA-16.250 fixed-angle rotor at 8900 × g at 4 °C for 20 min. The resulting supernatant was filtered through cheesecloth and then ultra-centrifuged in a Beckman Coulter Type 45Ti fixed-angle rotor at a speed of 20,000 × g at 4 °C for 1 h. The membrane pellets were divided into 20 aliquots. One aliquot was used immediately in the next step and the remaining aliquots were stored at −80 °C for future use. The membrane pellet fraction was subjected to a discontinuous sucrose gradient with steps of 0.15 mL 50%, 1.27 mL 36%, 1.27 mL 34%, 1.58 mL 32%, 1.58 mL 28%, 3.8 mL 25%, and 1.27 mL 14% sucrose. The sucrose gradient was then centrifuged in a Beckman Coulter SW 40Ti swinging-bucket rotor at 96,200 × g for 90 min. Bands at the interface of the 25 and 28% sucrose phases and at the interface of the 28 and 32% sucrose phases were confirmed to contain RyR1 by western blot. These bands were extracted from the sucrose gradient, diluted with dilution buffer (0.5 mM EDTA, 20 mM $Na_4O_7P_2$, 20 mM $NaH_2PO_4$, and 1 mM $MgCl_2$, pH 7.1) to 4 mL, and then ultra-centrifuged in a Beckman Coulter TLA 100.4 fixed-angle rotor at a speed of 40,000 × g at 4 °C for 20 min. The final membrane pellet was resuspended with 1 mL of dilution buffer. The tomograms were collected on the first-generation FEI Titan Krios (Thermo Fisher Scientific) at a nominal magnification of 81,000× corresponding to a pixel size of 1.7 Å per pixel in counting mode on a post-GIF Gatan K2 Summit electron detector. A total of 47 tomograms of SR vesicles containing RyR1 were collected using the script in Supplementary Note 1. An angular range of −60 to 60° was covered with a 3° step. The dose for the 0° image was 16 e$^-$/Å$^2$, and 2 e$^-$/Å$^2$ for the remaining images, with the target defocus set for between 3.5 and 4.5 µm.

**Image processing**. For the conventional processing, each tilt projection was aligned using MotionCor2[21]; and the average defocus was determined using Ctffind4[37] or Gctf[38]. The aligned projections were assembled into stacks and aligned using gold fiducials in Imod[22], and the resulting aligned tilt stacks were CTF-corrected using ctfphasefilp from Imod[26]. The generated CTF-corrected reconstructions were used for particle picking using the Dynamo catalogue tools[39]. Subtomogram averaging was performed using Dynamo[23], versions 1.281 and higher. For TMV, independent half-sets were generated by assigning different filaments into different sets; these sets were processed independently. For RyR1, independent half-sets were generated by dividing the particles to even and odd.

For the hybrid datasets, for the generation of the tilt series all the micrographs were normalized to have the same mean value of 128 and standard deviation of ~11. The "high-dose" images had a mean value of 128 and standard deviation of 4, which was 2.7 times lower, according to the relation $(\sigma_{HD})^2/(\sigma_{LD})^2 = e^-_{LD}/e^-_{HD}$. The generated stacks were processed according to the conventional StA processing workflow described above. After the conventional processing, the data were exported to Relion 3.0 using a Matlab script described in Supplementary Note 3 and a relion_refine refinement command was run on two independent half-sets in Relion[24,40]. Per-particle CTF correction was performed in cryoSPARC[41]. Tomograms were filtered by non-linear anisotropic diffusion[42] for display purposes. The description of the dyn2rel package, its parameters and an example of usage for the TMV dataset is presented in Supplementary Note 3.

Misalignments for Supplementary Fig. 1 was performed by adding random variables to the shifts and angles, which are required to align subtomograms to the average. The random variables had Gaussian distributions with zero mean values and standard deviations of 1, 3, and 5 for the small, moderate, and significant misalignments, respectively. For the precision of defocus determination measurements in Fig. 2c, 47 high-dose images from the RyR1 dataset were used. Gctf[38] was used to estimate the defocus and resolution, initially without restricting the frequencies used for the estimation. Then the higher frequencies up to 5, 10, 15, and 20 Å were excluded from calculation using the −resH option. The root mean defocus error was calculated by comparing the restricted results to the Gctf result with the full frequency range.

**Reporting summary**. Further information on experimental design is available in the Nature Research Reporting Summary linked to this paper.

## Data availability

The test hybrid datasets for TMV and RyR1 have been deposited to EMPIAR along with the alignment parameters for the tomograms, particle locations, and the alignment parameters for the particles in order to produce the hybrid map (EMPIAR-10393 for TMV and 10452 for RyR1). The maps are deposited to EMDB: TMV hStA map- EMD -10834, RyR1 hStA map - EMD-10840. Other data are available from the corresponding author upon request.

## Code availability

The SerialEM scripts HybridDoseSymmetricTomo and DuringTomoHybridSta are available in Supplementary Notes 1 and 2. The processing workflow is implemented in MATLAB (Mathworks).The documented scripts are presented in Supplementary Note 3, the dyn2rel code is available on Github (https://github.com/KudryashevLab/dyn2rel).

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

## Acknowledgements
We thank Deryck Mills for his expert electron microscopy support and the sample of TMV. We thank Dr. Kendra Leigh for critical reading of the manuscript, Wim Hagen for valuable discussions. The work was funded by the Sofja Kovalevskaja Award from the Alexander von Humboldt Foundation to M.K. R.S. is partially supported by the starter fellowship from SFB807 from the German Research Foundation. Y.Z. is partially supported by the IMPRES international student scholarship from the Max Planck Society. W.C. is supported by a fellowship from the China Scholarship Council. L.D. is supported by the Max Planck Society.

## Author contributions
Y.Z. modified the data collection script HybridDoseSymmetricTomo, prepared the TMV cryo samples, collected the data and contributed to data analysis. R.M.S. wrote the processing scripts and performed data analysis. L.D. collected the TMV dataset used for Fig. 3d–g. W.C. prepared the sample of RyR1, collected data and contributed to data processing for Fig. 4. M.K. designed and supervised the project, contributed to data analysis, acquired funding. M.K. wrote the manuscript with contributions from R.M.S., Y.Z., W.C., L.D.

## Competing interests
The authors declare no competing interests.
