## [Peer Review File · Nature Communications]

Reviewer #1 (Remarks to the Author):

In this paper, Sanchez, Zhang et al. present a hybrid method for structural determination of macromolecular complexes in native environments. Cryo-tomography and subtomogram averaging are used to identify particles and assign approximate positions and orientations, while single-particle methods are used to refine the orientation parameters of the same particles. To do this, a high dose image is collected at zero tilt, and lower dose images are appended to collect the rest of the tilt series.

The authors are able to convincingly show a significant improvement in the resolution and quality of the reconstructions.

While the approach is not original (as the authors comment in their discussion, a similar approach has been used in Tygress (Song et al, Nat Methods 2020)), the authors provide tools that are a useful addition to the set currently available for subtomogram averaging projects, and I support publication in Nat Comms.

Some points of discussion outlined below should be addressed before publication:

1. The authors have collected a zero tilt image at 15 e-/Å²s, and tilt series with an accumulated total dose of 95 e-/Å²s. It is unclear how the 95 e-/Å²s was determined as the best total dose. Recent subtomogram averaging studies have used higher accumulated doses, taking advantage of dose filtering. In my lab's experience, it is often possible to reach doses of 120-150 e-/Å²s before samples start to 'bubble' and become deformed. The best approach would probably be to determine the 'near-bubbling' dose for any given sample, reserve ~2 e-/Å²s for each tilted image and use the remaining available dose on the zero-tilt image. While in the discussion the authors comment on the possibility of changing the balance between dose in the zero versus tilted images, I wonder whether they should suggest a more systematic approach to maximise the dose on the zero tilt image, as suggested above. After all, the single particles that are going to be refined will be more noisy than 'traditional' single particles, as a variety of other structures (membranes, neighbouring proteins, etc.) will be overlapping along the projection direction, so maximising the dose of the zero tilt image seems to me like a good idea.

2. Particle z-height is used to adjust the value of defocus. One should be careful when doing this, as opposite handedness of the reconstructed tomogram will lead to inversion of the sign of the z-shift. Whether to add or subtract the z-height value from the average defocus when modifying the star file depends on the handedness, and can be directly related to which side of the tilt axis is closer to focus at positive or negative tilt. The authors should comment on this, and ideally they should explicitly introduce a flag in their scripts to account for a potential change in handedness due to different microscope settings or processing procedures.

3. The comparison in Figure 2 should be done between two zero-tilt images collected at different doses. As it stands now, the degradation in the CTF signal of the second image is due to a combination of lower dose, and tilt (introducing defocus gradient and slightly increasing thickness).

4. The authors currently mention their method might help with lower quality tomograms. It would be interesting to process the 15 e-/Å²s zero-tilt images of TMV following a pure single particle or helical reconstruction routine, to compare the results with their 5.2 Å refined map. Are the orientation and position parameters obtained with StA accurate enough, and the subsequent SPA refinement range generous enough to reach the best possible solution? If this is the case, then the resolutions of the two experiments should be comparable. Even if the authors don't feel they can do this analysis, it would be useful to comment on the necessity to reach a minimum level of accuracy with the tomography/subtomogram averaging part of the workflow to be able to successfully refine later.

Signed: Giulia Zanetti

Reviewer #2 (Remarks to the Author):

The manuscript from Sanchez et al. describes a new hybrid workflow to perform in situ structure determination for cryoEM. Here, the authors combine established sub-volume averaging procedures with single particle cryoEM approaches to achieve higher resolution particle maps. This hybrid method nicely combines the most outstanding merits from well established and widely used cryoEM software packages for cryoEM: SerialEM and IMOD can work seamlessly together to generate tomograms from tilt-series acquired based on the dose-symmetric scheme with a high throughput. Dynamo offers very versatile particle picking tools, user-friendly Matlab environment and GPU implemented subtomogram alignment procedure. Meanwhile, Relion has a very sophisticated particle refinement algorithm. It is very nice to see this hybrid method prioritize the high-resolution tomographic data throughout the whole pipeline from data acquisition to processing. This hStA method is a good example of using the tools collectively to gain new advantages for in situ structural studies. The two test samples nicely illustrate the potential of this method.

Major:

1. The zero degree image is the most crucial one that will largely determine the final resolution. However, the selected dose of 15e/A is not well explained: How was this determined as the optimal setting? This should be tested more rigorously, collecting datasets with different doses of the zero-degree images. What dose is clearly too little exposure, what is too much? How sample dependent is this optimal zero dose, and is based on trial-and-error? Or, is there a rule-of-thumb percentage of the total dosage to apply (15%)? This 15% of the total dose selection is only briefly mentioned in Line 217-219.

It would be very beneficial to include a more detailed guideline for choosing the high-dose and the reasoning behind would make the Hybrid StA more reader friendly, and potentially more user friendly.

2. The Refined Map is reconstructed solely from particles that are extracted from the zero-tilt projection. With this approach, the tomographic reconstruction and the StA alignment provide only the per-particle defocus values and the translational and rotational matrixes.

In this case, whether the zero-tilt can provide a sufficient coverage of the Fourier sampling of the targets of interest will become rather crucial for a successful 3D map reconstruction. This is not a concern for TMV dataset due to its helical symmetry, but for the RyR1 dataset it would be interesting to see more analysis on the angular distribution graphs of the Hybrid map.

Furthermore, a representative tomogram of RyR1 should be added to the supplementary for readers who are not overly familiar with the sample to get a better idea of the particle distribution and orientation.

3. It would be nice if you can comment on the outlook of using hStA on samples have a strong orientation preference (and little symmetry to save it). Would anisotropic resolution become the obstacle for getting a reasonable Hybrid map and hence the Refined map?

4. In the discussion, the authors acknowledge that other approaches also utilize similar methods and workflows to achieve hybrid cryoEM-StA. In this context, it would be appropriate to highlight the work from the Briggs group who, to my knowledge, pioneered this approach as early as 2014 (Bharat et al, PNAS June 3, 2014 111 (22) 8233-8238).

Minor:

Line 24-26: This sentence is both utterly important yet quite misleading.

The current sentence describes only the part of the single particle reconstruction process of the hStA. It is not obvious why collecting tilt series is at all necessary. It would be beneficial to

emphasize that the particle alignment information and the per-particle defocus determination is done via conventional StA?

Line 39: No obvious reason to keep the word "Futhermore";

Line 58: The values listed here may be typical, but many different electron doses and number of projections have been used by the diverse tomography labs. Maybe include the word 'typically' here (doses of over 200 e/A² and tilt series with >200 tilt images have been reported).

Line 69: At lower magnification, the particles appear smaller in a bigger field of view. In the paper, the authors state that this results in higher anisotropic magnification distortion. If I understand Grant & Grigorieff's paper correctly, magnification distortions are more problematic for large particles. Doesn't this mean that the particles at lower magnification would suffer the same or LESS from anisotropic magnification distortions? Also the amount of distortion is magnification depend, even on the same optics/microscope. Overall, the logic behind "using lower mag is at a cost of suffering more from anisotropic magnification" is not clear to me. More explanation would be beneficial.

Line 93: The word "micrographs" likely to lead to misunderstanding especially in tomography. A micrograph could mean either "a single frame within a movie stack that corresponds to a tilt angle (frame)" or "the 2D projection aligned and summed from a movie stack that corresponds to a tilt angle (tilt)". Presumably here it means the later one, it would be nice to avoid possible misunderstanding.

Line 94: Please specify in which software do you normalize mean value and adjust the standard deviation.

It would be nice to include supplementary information that illustrates how these data processing steps are done, and how this may affect tomogram reconstruction. Is this processing step mandatory for all hStA tilt series (even process for conventional StA workflow)?

Line 99: Please specify how do you determine defocus and correct CTF in the StA workflow. It is later listed in the method, but since the workflow is introduced here, it would also be convenient to include it here.

Line 128: "2.2 A/pixel in counting mode". Specify recorded in super resolution mode is perhaps more important here.

Line 130: A bit ambiguous to use the term "particles" here. Does it mean the manually picked TMV fibrils? It cannot be the asymmetric units, right?

Line 134: It is unclear in the text or in the figure legend what the gray and blue curves in Fig.2C are and where the data came from.

With best wishes,
Wen Yang and Ariane Briegel

Dear Reviewers,

Thank you very much for the interest in our manuscript and the constructive comments. We followed them and updated the manuscript and the code, please find or point-by-point response below.

Reviewer #1 (Remarks to the Author):

In this paper, Sanchez, Zhang et al. present a hybrid method for structural determination of macromolecular complexes in native environments. Cryo-tomography and subtomogram averaging are used to identify particles and assign approximate positions and orientations, while single-particle methods are used to refine the orientation parameters of the same particles. To do this, a high dose image is collected at zero tilt, and lower dose images are appended to collect the rest of the tilt series.

The authors are able to convincingly show a significant improvement in the resolution and quality of the reconstructions.

While the approach is not original (as the authors comment in their discussion, a similar approach has been used in Tygress (Song et al, Nat Methods 2020)), the authors provide tools that are a useful addition to the set currently available for subtomogram averaging projects, and I support publication in Nat Comms.

Some points of discussion outlined below should be addressed before publication:

1. The authors have collected a zero tilt image at 15 e-/Å²s, and tilt series with an accumulated total dose of 95 e-/Å²s. It is unclear how the 95 e-/Å²s was determined as the best total dose. Recent subtomogram averaging studies have used higher accumulated doses, taking advantage of dose filtering. In my lab's experience, it is often possible to reach doses of 120-150 e-/Å²s before samples start to 'bubble' and become deformed. The best approach would probably be to determine the 'near-bubbling' dose for any given sample, reserve ~2 e-/Å²s for each tilted image and use the remaining available dose on the zero-tilt image. While in the discussion the authors comment on the possibility of changing the balance between dose in the zero versus tilted images, I wonder whether they should suggest a more systematic approach to maximise the dose on the zero tilt image, as suggested above. After all, the single particles that are going to be refined will be more noisy than 'traditional' single particles, as a variety of other structures (membranes, neighbouring proteins, etc.) will be overlapping along the projection direction, so maximising the dose of the zero tilt image seems to me like a good idea.

We indeed followed the outlined logic in our experimental design. In our TMV experiment we wanted to maintain the 11 Å signal which is needed to align the helices to each other. This peak is seen in Figure 3C and in our experience if the individual subtomograms do not have this information, the alignment does not converge. Looking into the dose-dependent resolution decay outlined in (Grant & Grigorieff, 2015a), we decided to use total electron doses slightly under 100 e-/Å², as at these doses the ~10 Å signal is still present in the data. As the reviewer pointed out, a dose of ~2 e-/Å² per projection is needed to produce a good tomogram. If the

sample can tolerate higher electron dose or needs less projections to allow subtomograms to be aligned, the additional dose indeed should be used for the high dose exposure.

We outlined these considerations in a new paragraph of the manuscript (lines 245-257)

2. Particle z-height is used to adjust the value of defocus. One should be careful when doing this, as opposite handedness of the reconstructed tomogram will lead to inversion of the sign of the z-shift. Whether to add or subtract the z-height value from the average defocus when modifying the star file depends on the handedness, and can be directly related to which side of the tilt axis is closer to focus at positive or negative tilt. The authors should comment on this, and ideally they should explicitly introduce a flag in their scripts to account for a potential change in handedness due to different microscope settings or processing procedures.

Thank you for an insightful comment, we added the “direction of the defocus” to the code (on GitHub) and updated the user manual (lines 1009-1012). Our suggestion is to try both options if the user is not sure about the handedness.

```
% exporter.inv_dz = true;
```

3. The comparison in Figure 2 should be done between two zero-tilt images collected at different doses. As it stands now, the degradation in the CTF signal of the second image is due to a combination of lower dose, and tilt (introducing defocus gradient and slightly increasing thickness).

We agree with the reviewer. We performed summing-up of the high-dose images either fully (15 e-/Å²s) or partially (first 2 e-/Å²s) and compared the resulting images. We updated the figure and the description accordingly (Figure 2, figure legend and lines 141-142).

4. The authors currently mention their method might help with lower quality tomograms. It would be interesting to process the 15 e-/Å²s zero-tilt images of TMV following a pure single particle or helical reconstruction routine, to compare the results with their 5.2 Å refined map. Are the orientation and position parameters obtained with StA accurate enough, and the subsequent SPA refinement range generous enough to reach the best possible solution? If this is the case, then the resolutions of the two experiments should be comparable. Even if the authors don't feel they can do this analysis, it would be useful to comment on the necessity to reach a minimum level of accuracy with the tomography/subtomogram averaging part of the workflow to be able to successfully refine later.

We thank the reviewer for the suggestion, we performed the calculation. We perturbed the alignment parameters for subtomogram averaging of TMV with the controlled amount of noise (3 levels). This led to significant measurable degradation of resolution, which we could recover by running the hybrid refinement. We describe this in the text (lines 191-196), in the methods (lines 375-378) and added a supplementary figure describing the results. The results of these calculations support

our conclusions about slightly relaxing the quality requirements for tomograms, in particularly the low dose images (in the last paragraph of Discussion).

The ab-initio helical reconstruction from only the high-dose images resulted in 4.4 Å resolution, reaching Nyquist, however it has been shown before that it is rather straightforward to obtain TMV reconstructions by helical reconstruction and we don't see novelty in this reconstruction.

We furthermore further explored the possibility of refining the defocus values in 2D on "per particle basis" and also reached the Nyquist frequency of 4.4 Å. We added a cautious statement in the discussion that such refinement could be beneficial for some samples (lines 239-241).

Signed: Giulia Zanetti

Reviewer #2 (Remarks to the Author):

The manuscript from Sanchez et al. describes a new hybrid workflow to perform in situ structure determination for cryoEM. Here, the authors combine established sub-volume averaging procedures with single particle cryoEM approaches to achieve higher resolution particle maps.

This hybrid method nicely combines the most outstanding merits from well established and widely used cryoEM software packages for cryoEM: SerialEM and IMOD can work seamlessly together to generate tomograms from tilt-series acquired based on the dose-symmetric scheme with a high throughput. Dynamo offers very versatile particle picking tools, user-friendly Matlab environment and GPU implemented subtomogram alignment procedure. Meanwhile, Relion has a very sophisticated particle refinement algorithm. It is very nice to see this hybrid method prioritize the high-resolution tomographic data throughout the whole pipeline from data acquisition to processing. This hStA method is a good example of using the tools collectively to gain new advantages for in situ structural studies. The two test samples nicely illustrate the potential of this method.

Major:

1. The zero degree image is the most crucial one that will largely determine the final resolution. However, the selected dose of 15e/Å is not well explained: How was this determined as the optimal setting? This should be tested more rigorously, collecting datasets with different doses of the zero-degree images. What dose is clearly too little exposure, what is too much? How sample dependent is this optimal zero dose, and is based on trial-and-error? Or, is there a rule-of-thumb percentage of the total dosage to apply (15%)? This 15% of the total dose selection is only briefly mentioned in Line 217-219.

It would be very beneficial to include a more detailed guideline for choosing the high-dose and the reasoning behind would make the Hybrid StA more reader friendly, and potentially more user friendly.

Thank you very much for the review and the constructive comment.

Our motivation to use sub-100 e⁻/Å² was to still maintain the useful structural information in the tomograms. It is surprisingly difficult to perform subtomogram

averaging of TMV helices and it requires the 11 Å signal to be present in the data. We therefore aimed at preserving it based on *Grant and Grigorieff, eLife, 2015*, and exposing each of the 40 projections to $2 \text{ e}^-/\text{Å}^2$ to provide significant contrast and enough Thon rings for CTF correction.

For the RyR1 we used an exposure on the relative low total electron dose to show that $16 \text{ e}^-/\text{Å}^2$ is sufficient to refine the alignment of the particle; the maximal dose per tilt series is sample-specific and is defined by the tolerance to the electron dose. We added an additional paragraph discussing the distribution of electron dose (lines 245-257)

2. The Refined Map is reconstructed solely from particles that are extracted from the zero-tilt projection. With this approach, the tomographic reconstruction and the StA alignment provide only the per-particle defocus values and the translational and rotational matrixes.

In this case, whether the zero-tilt can provide a sufficient coverage of the Fourier sampling of the targets of interest will become rather crucial for a successful 3D map reconstruction. This is not a concern for TMV dataset due to its helical symmetry, but for the RyR1 dataset it would be interesting to see more analysis on the angular distribution graphs of the Hybrid map.

Furthermore, a representative tomogram of RyR1 should be added to the supplementary for readers who are not overly familiar with the sample to get a better idea of the particle distribution and orientation.

Indeed, a preferred orientation might be a critical limitation for obtaining a well-sampled map. In both our cases we obtained near-isotropic particle distributions. For the more challenging case of RyR1 we added a Supplementary Figure 2 demonstrating side- and top-views and their distribution in the dataset as well as a video displaying a representative tomogram (Supplementary Movie 1) and references to them in the text (lines 203 and 205-207).

3. It would be nice if you can comment on the outlook of using hStA on samples have a strong orientation preference (and little symmetry to save it). Would anisotropic resolution become the obstacle for getting a reasonable Hybrid map and hence the Refined map?

Yes, lack of symmetry and preferred orientation indeed will result in an anisotropic resolution of the hStA map. For such cases, we suggested recording a high-dose image at a ~20-30 degree tilt. For such data collection, we added an additional script to the (new) supplemental note 2 which could be used alongside with the standard SerialEM tomography functionality (with the tilt series setup, instead of using the script for data collection). In this setup a unidirectional tilt series could be recorded, and a high-dose image can be acquired at i.e 21 degrees (`TargetHighDoseAngle = 21`). This solution is optimized for robustness and performance, however comes at a price of recording several low-dose images prior to acquiring the high-dose image. This script could be also used with setting the target angle to 0.

In the case of tilted data collection, image processing will work the same way. For processing the user needs to update the number of the high-dose image in the stack, value *HD_ix*, we added the option to the Supplementary Note 3 (line 983-984).

We added discussion on this subject (lines 257-262).

4. In the discussion, the authors acknowledge that other approaches also utilize similar methods and workflows to achieve hybrid cryoEM-StA. In this context, it would be appropriate to highlight the work from the Briggs group who, to my knowledge, pioneered this approach as early as 2014 (Bharat et al, PNAS June 3, 2014 111 (22) 8233-8238).

Thank you, we added the even earlier work with the same concept - Bharat, Nature, 2012, and partially re-written the corresponding paragraph in the discussion (lines 263-266)

Minor:

Line 24-26: This sentence is both utterly important yet quite misleading. The current sentence describes only the part of the single particle reconstruction process of the hStA. It is not obvious why collecting tilt series is at all necessary. It would be beneficial to emphasize that the particle alignment information and the per-particle defocus determination is done via conventional StA?

In the case of TMV or purified ribosomes it is not needed to perform record tomograms at all, and we used it as a proof-of principle sample. The real need for tomography is illustrated by the RyR1 example where we failed to pick particles by templates in 2D from high dose images using gautomatch (data not shown). We updated the abstract focusing on the need to pick particles from the tomograms in 3D (lines 24-27). We Furthermore attempted to discard the angular information and to perform 360 degree-searches. In such cases we could not obtain high-resolution reconstructions of RyR1, we now write about this in lines 209-212.

We explored the possibility of refining the defocus values in 2D on “per particle basis” for the TMV dataset and reached the Nyquist frequency of 4.4 Å. We added a cautious statement in the discussion that such refinement could be beneficial for some samples (lines 239-241). Per-particle refinement of CTF is recommended after reaching ~5 Å in resolution and we have not attempted it for the RyR1 dataset.

Line 39: No obvious reason to keep the word “Futhermore”;

Agreed, thank you.

Line 58: The values listed here may be typical, but many different electron doses and number of projections have been used by the diverse tomography labs. Maybe include the word ‘typically’ here (doses of over 200 e/A² and tilt series with >200 tilt images have been reported).

Thank you very much, we followed your suggestion.

Line 69: At lower magnification, the particles appear smaller in a bigger field of

view. In the paper, the authors state that this results in higher anisotropic magnification distortion. If I understand Grant & Grigorieff's paper correctly, magnification distortions are more problematic for large particles. Doesn't this mean that the particles at lower magnification would suffer the same or LESS from anisotropic magnification distortions? Also the amount of distortion is magnification depend, even on the same optics/microscope. Overall, the logic behind "using lower mag is at a cost of suffering more from anisotropic magnification" is not clear to me. More explanation would be beneficial.

Thank you, the problem with anisotropic magnification is that it elongates the micrographs by up to 2.7% making the tomographic reconstruction incoherent. At the edge of a 4k image distortion could be up to ~50 pixels which will make the tomograms much worse especially at the periphery. Magnification anisotropy depends on the used magnification and it is hard to calibrate at pixels sizes which do not allow observing i.e. gold diffraction (~ 2Å) or graphite (3.4 Å).

We expanded our introduction on this issue (lines 71-77).

Line 93: The word "micrographs" likely to lead to misunderstanding especially in tomography. A micrograph could mean either "a single frame within a movie stack that corresponds to a tilt angle (frame)" or "the 2D projection aligned and summed from a movie stack that corresponds to a tilt angle (tilt)". Presumably here it means the later one, it would be nice to avoid possible misunderstanding.

Thank you, we now defined the "micrographs" as the result of the motion correction procedure (line 50) and paid particular attention to the definition further in the text.

Line 94: Please specify in which software do you normalize mean value and adjust the standard deviation.

It would be nice to include supplementary information that illustrates how these data processing steps are done, and how this may affect tomogram reconstruction. Is this processing step mandatory for all hStA tilt series (even process for conventional StA workflow)?

We now provide a section on preprocessing of input stacks in the Supplementary Note 3 (former Supplementary Note 2). We use the Imod's command *newstack* and provide the commands.

Line 99: Please specify how do you determine defocus and correct CTF in the StA workflow. It is later listed in the method, but since the workflow is introduced here, it would also be convenient to include it here.

We outlined two options for CTF determination in our Supplementary Note 3 (former Supplementary Note 2), one of which is a detailed script (lines 836-894 and below).

Line 128: "2.2 Å/pixel in counting mode". Specify recorded in super resolution mode is perhaps more important here.

Thank you, we followed your suggestion.

Line 130: A bit ambiguous to use the term “particles” here. Does it mean the manually picked TMV fibrils? It cannot be the asymmetric units, right?

These are indeed asymmetric units, we updated the description.

Line 134: It is unclear in the text or in the figure legend what the gray and blue curves in Fig.2C are and where the data came from.

Thank you, we updated the figure legend and made a reference to the Method section in the text.

**With best wishes,
Wen Yang and Ariane Briegel**

REVIEWERS' COMMENTS:

Reviewer #1 (Remarks to the Author):

I am satisfied with the authors' responses to my comments and I recommend publication of the manuscript in its current form.

Reviewer #2 (Remarks to the Author):

We are content with the revised version of the manuscript.